# Risk Factors of Ixekizumab-Induced Injection Site Reactions in Patients with Psoriatic Diseases: Report from a Single Medical Center

**DOI:** 10.3390/biomedicines11061718

**Published:** 2023-06-15

**Authors:** I-Heng Chiu, Tsen-Fang Tsai

**Affiliations:** 1Department of Dermatology, National Taiwan University Hospital, Taipei 100, Taiwan; heng821993@gmail.com; 2Department of Dermatology, College of Medicine, National Taiwan University, Taipei 100, Taiwan

**Keywords:** injection site reactions, ixekizumab, psoriasis, biologics

## Abstract

Ixekizumab (Taltz^®^) is a humanized anti-IL-17A monoclonal antibody approved for the treatment of various inflammatory diseases including psoriasis and psoriatic arthritis. Despite the favorable efficacy and safety, ixekizumab is also known for its high incidence of injection site reactions (ISRs), ranging from 6% to 55% in different studies according to different definitions and studied population. However, specific risk factors for ixekizumab-induced injection site reactions in patients with psoriatic diseases had not been well studied. In this retrospective study, we found that overweight or obesity might be a protective predictor for the occurrence of ixekizumab-induced ISRs in patients with psoriatic disease. Meanwhile, having a positive family history of psoriasis might be a potential risk factor. Last but not least, patients with diarrhea following ixekizumab injection were associated with a higher risk of developing ISRs. Future high-quality studies with larger samples are warranted to verify the relationship.

## 1. Introduction

Psoriasis is a chronic skin disease characterized by immune-mediated inflammation. It affects approximately 2% of the global population and is characterized by the presence of distinct red, scaly plaques that are well demarcated from the surrounding unaffected areas [1,2]. The impact of psoriasis on individuals’ quality of life can be profound, leading to devastating consequences and stigmatization [3]. There are several approved targeted systemic and biologic therapies available for the management of moderate-to-severe psoriasis. Ixekizumab (Taltz®) is a humanized anti-IL-17A monoclonal antibody that has received approval from the Food and Drug Administration (FDA) for the treatment of moderate to severe plaque psoriasis in patients who are candidates for systemic therapy or phototherapy, active psoriatic arthritis, active ankylosing spondylitis, and active non-radiographic axial spondyloarthritis with objective signs of inflammation [4]. Clinical trials have demonstrated its favorable efficacy and safety profiles, with significant improvements in the symptoms and quality of life for patients [5]. However, it is important to note that occasional adverse events have been reported with the use of ixekizumab. The most common adverse events include nasopharyngitis, upper respiratory tract infection, injection site reactions (ISRs), and headache [6]. In the dermatological aspect, adverse events such as paradoxical psoriasis and atopic-like eczema have been reported. Rarely, the development of neutrophilic dermatoses, hypersensitivity reactions, lichenoid eruptions, vasculopathies, granulomatous diseases, lupus-like reactions, hair disorders, and vitiligo have also been reported in association with the use of ixekizumab [7,8,9]. Close monitoring and awareness of these potential adverse events are necessary during treatment with ixekizumab. Additionally, it is noteworthy that ixekizumab has a relatively high incidence of ISRs compared to other biologics within the same class [10]. The reported incidence of ISRs with ixekizumab varies between 6% and 55% in different studies, likely due to variations in definitions and studied populations [11,12]. ISRs are predominantly mild to moderate in severity and rarely lead to discontinuation of ixekizumab. Most of the cases were reported in the first 2 weeks after ixekizumab administration and resolved spontaneously within a few days. The incidence of ISRs usually decreases over time after subsequent injections [13].

The definition of ISRs varies in different studies and can refer to various reactions. While it is usually defined as swelling, erythema, pruritus, or pain around the site of injection, other terms may also be used to indicate different types of reactions [14]. ISRs can be classified as physical, irritant, or allergic depending on the underlying cause, such as needle puncture or injection technique, ingredients of the injected solution, or immediate and delayed allergic reactions [12]. Typically, the onset of erythema after drug administration is delayed, while pain develops earlier [13]. Histologically, perivascular lymphocytic infiltrate with eosinophils and mucin deposits have been reported [15].

In the management of ISRs, several remedies have been proposed. One approach is to avoid injecting in the same site repeatedly and maintain a distance of at least 3 cm from the previous injection site [16]. This helps to prevent further irritation and inflammation in the area. Applying cold therapy, such as ice packing or a wet towel, to the injection site for 10 to 15 min after the injection can help alleviate symptoms and reduce inflammation by numbing the area and constricting blood vessels [16]. Antihistamines can be considered to provide symptomatic relief by reducing itching and discomfort associated with ISRs [17]. Over-the-counter analgesics, such as acetaminophen or ibuprofen, can be used for pain relief. Topical corticosteroids may also be used to mitigate inflammation, reducing redness, swelling, and itching [5,17].

In our previous study, 44% of patients with erythrodermic psoriasis developed ixekizumab-induced ISRs [18], which was slightly higher than in pivotal studies of ixekizumab for psoriasis vulgaris. The objective of this study is to report the common risk factors associated with ixekizumab-induced ISRs and assess the overall incidence of these reactions in patients with psoriatic diseases.

## 2. Materials and Methods

Consecutive patients with psoriatic diseases who had received ixekizumab injection in the dermatologic outpatient clinic of our hospital between December 2015 and October 2022 were enrolled for analysis. The baseline body mass index (BMI) categories were classified according to the Ministry of Health and Welfare of Taiwan, defining obesity as BMI ≥ 27 kg/m^2^, overweight as 24 ≤ BMI < 27 kg/m^2^, normal as 18.5 < BMI < 24 kg/m^2^, and underweight as BMI < 18.5 kg/m^2^ using local statistic results [19]. The baseline severity of psoriasis was measured using the psoriasis area severity index (PASI) score and the percentage of body surface area (BSA) involvement. All of the PASI evaluation was assessed by the same physician. The occurrence of ISRs and diarrhea were routinely documented in each visit during the period of ixekizumab administration. We defined ISRs in our study as the presence of injection site swelling, with or without erythema or pain. Diarrhea after injection was inquired about orally, following the definition of the passage of three or more loose or liquid stools per day.

To assess the association between the occurrence of ISRs and potential risk factors, chi-squared test, Fisher’s exact test, univariate and multivariate logistic regression analyses were performed using SPSS version 16. The parametric data were presented as mean ± standard deviation.

## 3. Results

In total, 116 patients with psoriatic diseases were included in the study. The baseline demographic data are summarized in Table 1. The average age of the population was 51.2 ± 13.0 years, and the mean age of psoriasis onset was 29.6 ± 15.1 years. Most patients were male (73.3%), with a mean body weight of 78.1 ± 16.2 kg. The average baseline PASI score was 12.0 ± 8.8, and the average BSA was 17.8% ± 22.8%. A history of erythrodermic change was reported in 15.6% of patients, and 32.8% had a positive family history of psoriasis. Psoriatic arthritis was diagnosed in 63.8% of patients. The most commonly reported comorbidity was hypertension (42.2%), followed by diabetes mellitus (22.4%). A smaller proportion of patients reported having cardiovascular disease (3.4%) and hepatitis B virus infection (8.6%). Prior to ixekizumab treatment, most patients had received acitretin (70.7%), methotrexate (81.9%), phototherapy (78.4%), and had a history of prior biologics exposure (81.9%), ranging from 1 agent to 7 agents.

Figure 1 presents a clinical photograph showing ISRs observed in one of our patients. The study found that 34.5% of psoriasis patients who received ixekizumab developed ISRs, which persisted for an average duration of 2.6 ± 1.1 days. Another example demonstrating the progression of ISRs over a consecutive 3-day period following ixekizumab administration was depicted in Figure 2. The series of clinical photographs captured the evolving nature of ISRs in the patient, providing visual insights into the temporal changes associated with the reaction. Furthermore, 16.4% of patients experienced diarrhea after receiving ixekizumab, with an average duration of 2.3 ± 1.1 days. None of the subjects discontinued ixekizumab due to ISRs. By week 12, the mean PASI score improved to 2.7 ± 4.3, and the average BSA involvement was 2.4% ± 9.0%. At week 12, 69.8%, 44.8%, and 30.2% of patients achieved PASI 75, PASI 90, and PASI 100 response, respectively (as shown in Table 2).

Table 3 and Table 4 present the univariate and multivariate analyses of factors associated with ISRs, respectively. In the univariate analysis, significant factors included the patient’s age, onset age of PsO, family history of PsO, BMI ≥ 24 (indicating overweight or obesity), and diarrhea after injection. Of these factors, family history of PsO, BMI ≥ 24, and diarrhea after injection were the only variables that reached statistical significance in the multivariate analysis, with odds ratios of 4.61, 0.10, and 71.98, respectively. Notably, erythrodermic psoriasis was not found to be a risk factor for ixekizumab-induced ISRs. Finally, it is worth mentioning that the development of ISRs is not related to the treatment response of ixekizumab.

## 4. Discussion

An integrated database composed of three randomized trials in 2017 demonstrated a trend suggesting that a heavier body weight might be a protective factor against ixekizumab-induced ISRs in patients with plaque psoriasis [20]. According to the official package insert of ixekizumab from the European Medicines Agency, a study showed that ixekizumab-induced ISRs occurred in 25% of cases weighing less than 60 kgs who were diagnosed with plaque psoriasis, while the incidence was lower at 14% in cases weighing above 60 kgs [21]. These findings were consistent not only in patients with plaque psoriasis but also in those with psoriatic arthritis (24% in patients with body weight < 100 kg versus 13% in patients with body weight ≥ 100 kg) or axial spondyloarthritis (14% in patients with body weight < 100 kg versus 9% in patients with body weight ≥ 100 kg) [21]. However, in this observational study, no significant relationship was found between body weight and ixekizumab-induced ISRs. However, it is worth noting that a BMI of ≥24, which indicates overweight or obesity according to the Ministry of Health and Welfare of Taiwan, was identified as an independent protective factor against ixekizumab-induced ISRs. We postulated that this association may be related to differences in subcutaneous fat thickness between normal-weight and obese patients. Additionally, patients with a positive family history of psoriasis may have a higher risk of developing ISRs after receiving ixekizumab. This finding is consistent with previous reports indicating that a positive family history of psoriasis may reduce quality of life [22] and generate more treatment-related adverse events [23] in patients with psoriasis.

Although this was not proven in the multivariate analysis, younger patients or patients with earlier onset of psoriasis may have a higher risk of ixekizumab-induced ISRs. Previous studies have shown that the sensitization potential may decrease with age in patients with contact dermatitis [24,25]. We assume that this concept may also be applied to psoriasis patients who develop ISRs.

Regarding the aspect of post-injection diarrhea, it is well-known that IL-17 contributes to epithelial and mucosal protection [26]. Previous studies have reported an association between IL-17 blockade and profound gut microbiome shifts, as well as exacerbation of inflammatory bowel disease or gastrointestinal events [27]. Interestingly, patients who developed ixekizumab-induced diarrhea had an increased risk of ISRs. While the precise pathogenesis remains uncertain, we propose some possible explanations besides the nocebo effect. First, altered gut microbiome by IL-17 inhibitors may cause dysregulation of the gut–skin axis [28], interfering with the modulation of cutaneous immune response to various stimuli, finally leading to the occurrence of ISRs. Second, ixekizumab is commonly injected into the abdomen. ISRs and inflammation in the abdomen might potentially cause irritation or hypermotility of the bowel. Notably, the severity of diarrhea during ixekizumab treatment decreased over time in our patients, similar to the ISRs [29].

The incidence of ixekizumab-induced ISRs and diarrhea has been found to vary widely across different studies [11,12]. We believe that there may be an underestimation of the incidence if physicians in the clinic do not routinely inquire about and document these adverse events, as they are often mild and self-limiting. Furthermore, patients may have their own subjective definitions of diarrhea, which may lead to a risk of misestimation.

The ixekizumab solution contains additional ingredients, including citrate, polysorbate, sodium chloride, sodium citrate dihydrate, and water for injection [30]. While citric acid is a commonly used ingredient in pharmaceuticals, it has been known to cause injection site pain by activating the acid-sensing ion channel 1 [31]. Previous studies have shown that solutions containing citrate as a buffer can cause more pain after subcutaneous injection than those without [32]. Additionally, a citrate-free formulation of adalimumab has been shown to induce significantly less injection site pain than the original formulation [33], which could further support this hypothesis. A new citrate-free formulation of ixekizumab has been proven to be bioequivalent, with an 86% decrease in visual analog scale of pain and the same safety profile as the original commercial formulation [34].

This study has several limitations that should be acknowledged. Firstly, its retrospective and uncontrolled design may have introduced bias into the results, potentially affecting the generalizability of the findings. Secondly, the assessment of outcomes, such as ISRs and diarrhea, relied on self-reporting by patients during clinic visits, without direct visualization or objective measurements. By the time the patients returned to the clinic, the ISRs had all resolved. This introduces the possibility of recall bias or subjective interpretation. Thirdly, the study did not analyze the incidences of ISRs associated with each dose of ixekizumab, and the relationship between the dosing interval of ixekizumab and the incidence of ISRs was not investigated. This limits our understanding of potential dose-dependent effects. Lastly, the lack of comprehensive and detailed data regarding the characteristics of ISRs or diarrhea may have restricted the depth and accuracy of the findings. Future studies with more rigorous designs and larger sample sizes are needed to address these limitations and provide more robust evidence on the topic.

## 5. Conclusions

ISRs can occur in up to 50% of psoriasis patients receiving ixekizumab. Fortunately, most cases are mild in severity and resolve spontaneously, rarely requiring discontinuation of ixekizumab treatment. To alleviate ISRs, various approaches can be considered, including ice packing, the use of over-the-counter anti-histamines, analgesics, and topical steroids. It is also important to rotate between the injection sites to prevent the recurrence of ISRs. Overweight or obesity may be protective against ixekizumab-induced ISRs in patients with psoriatic diseases, while a positive family history of psoriasis may be a potential risk factor. Both of these factors are easily available in clinical practice. The development of ISRs is not related to the treatment response of ixekizumab. Additionally, psoriasis patients who experience ixekizumab-induced diarrhea are associated with a higher risk of developing ISRs. Future high-quality studies with larger participant samples are needed to verify this relationship.

## Figures and Tables

**Figure 1 biomedicines-11-01718-f001:**
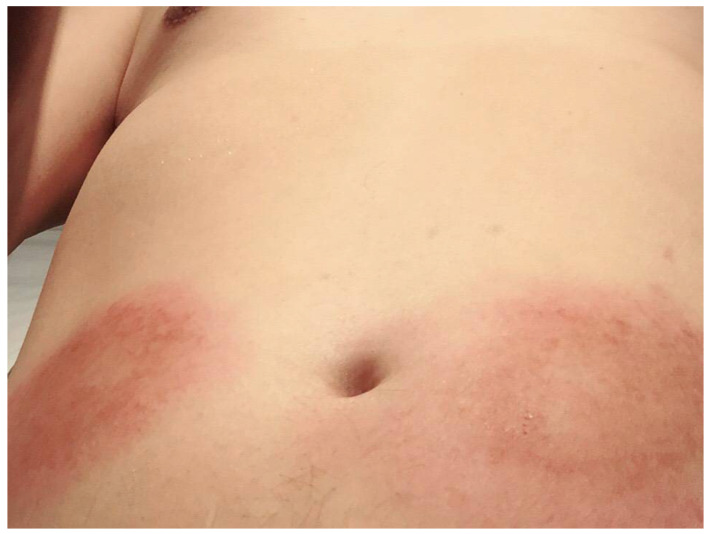
One example of ixekizumab-induced ISRs on the patient’s abdomen during the initial injection, with a total of two injections of 80 mg/pen, one administered on each side. Subsequently, no further ISRs occurred following his twelfth injection.

**Figure 2 biomedicines-11-01718-f002:**
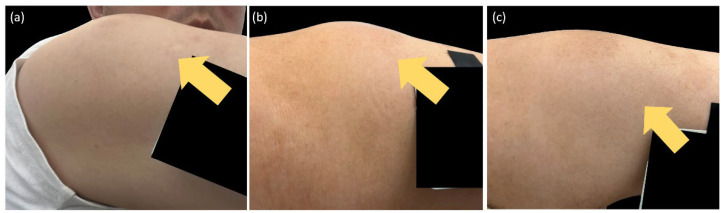
Progression of ISRs on the right upper arm (yellow arrows) over a period of 3 consecutive days following ixekizumab administration. (**a**) day 1. (**b**) day 2. (**c**) day 3. The patient has experienced ixekizumab-induced ISRs consistently from the first injection up to the present, including the most recent seventeenth injection.

**Table 1 biomedicines-11-01718-t001:** Baseline demographics and clinical characteristics.

Characteristics (N = 116)		
Age (year, mean ± SD)	51.2	13.0
Gender (male)	85	73.3%
Height (cm, mean ± SD)	167.4	8.11
Weight (kg, mean ± SD)	78.1	16.2
Onset age of PsO (year, mean ± SD)	29.6	15.1
FH of PsO	38	32.8%
Habit		
Smoking	46	39.7%
Alcohol	25	21.6%
BMI		
Underweight	0	0%
Normal	24	20.7%
Overweight	32	27.6%
Obesity	60	51.7%
Baseline severity of PsO		
PASI (mean ± SD)	12.0	8.8
BSA (mean ± SD)	17.8	22.8
Underlying diseases		
Erythrodermic PsO	18	15.6%
PsA	74	63.8%
HTN	49	42.2%
DM	26	22.4%
CVD	4	3.4%
HBV	10	8.6%
IGRA	21	18.1%
Prior PsO treatments		
Acitretin	82	70.7%
MTX	95	81.9%
CsA	29	25.0%
Phototherapy	91	78.4%
Prior Biologic agents	95	81.9%
1 agent	27	23.3%
2 agents	24	20.7%
3 agents	17	14.7%
4 agents	14	12.1%
5 agents	8	6.9%
6 agents	5	3.4%
7 agents	0	0%
8 agents	1	0.9%

PsO, psoriasis; FH, family history; BMI, body mass index (the weight in kilograms divided by the square of the height in meters); PASI, psoriasis area and severity index; BSA, body surface area; PsA, psoriatic arthritis; HTN, hypertension; DM, diabetes mellitus; CVD, cardiovascular disease; HBV, hepatitis B virus; IGRA, interferon-gamma release assay; MTX, methotrexate; CsA, cyclosporine; SD, standard deviation.

**Table 2 biomedicines-11-01718-t002:** Events of special interests from ixekizumab treatment.

Event Type (N = 116)		
ISRs		
Incidence	40	34.5%
Duration (day, mean ± SD)	2.6	1.1
Diarrhea		
Incidence	19	16.4%
Duration (day, mean ± SD)	2.3	1.1
Severity of PsO at week 12		
PASI (mean ± SD)	2.7	4.3
BSA (mean ± SD)	2.4	9.0
PASI response at week 12		
PASI 75	81	69.8%
PASI 90	52	44.8%
PASI 100	35	30.2%

ISRs, injection site reactions; PsO, psoriasis; PASI, psoriasis area and severity index; BSA, body surface area; SD, standard deviation.

**Table 3 biomedicines-11-01718-t003:** Univariate analysis of factors associated with ixekizumab-induced injection site reactions.

Characteristics	OR	(95.0% C.I.)	*p* Value
Age	0.95	(0.92–0.98)	0.002
Height	1.01	(0.96–1.06)	0.70
Weight	1.00	(0.98–1.02)	0.89
Onset age of PsO	0.96	(0.93–0.99)	0.003
Gender (male)	0.54	(0.23–1.25)	0.14
Family history of PsO	2.71	(1.21–6.10)	0.01
Habit			
Smoking	0.87	(0.40–1.91)	0.73
Alcohol	0.87	(0.34–2.23)	0.77
BMI ≥ 24	0.35	(0.14–0.88)	0.02
Baseline PASI	1.00	(0.95–1.04)	0.87
Underlying diseases			
Erythrodermic PsO	1.25	(0.45–3.53)	0.67
PsA	1.08	(0.49–2.41)	0.84
HTN	0.74	(0.34–1.62)	0.45
DM	0.64	(0.24–1.67)	0.36
CVD	0.62	(0.06–6.20)	1.00
HBV	0.45	(0.09–2.22)	0.49
IGRA	0.54	(0.18–1.59)	0.26
Prior PsO treatments			
Acitretin	1.69	(0.70–4.08)	0.24
MTX	1.39	(0.50–3.93)	0.53
CsA	1.81	(0.76–4.27)	0.18
Phototherapy	1.89	(0.69–5.19)	0.21
Prior biologic agents	1.28	(0.45–3.63)	0.64
Diarrhea after injection	7.65	(2.51–23.33)	<0.001
PASI response at week 12			
PASI 75	1.22	(0.52–2.84)	0.65
PASI 90	1.18	(0.55–2.54)	0.68
PASI 100	1.41	(0.62–3.21)	0.41

PsO, psoriasis; BMI, body mass index; PASI, psoriasis area and severity index; PsA, psoriatic arthritis; HTN, hypertension; DM, diabetes mellitus; CVD, cardiovascular disease; HBV, hepatitis B virus; IGRA, interferon-gamma release assay; MTX, methotrexate; CsA, cyclosporine.

**Table 4 biomedicines-11-01718-t004:** Multivariate analysis of factors associated with ixekizumab-induced injection site reactions.

Characteristics	OR	(95.0% C.I.)	*p* Value
Age	0.95	(0.88–1.02)	0.14
Height	1.08	(0.94–1.25)	0.27
Weight	0.98	(0.92–1.04)	0.48
Onset age of PsO	0.98	(0.91–1.05)	0.50
Gender (male)	0.14	(0.01–1.40)	0.09
Family history of PsO	4.61	(1.25–17.09)	0.02
Habit			
Smoking	1.90	(0.47–7.70)	0.37
Alcohol	0.47	(0.09–2.49)	0.37
BMI ≥ 24	0.10	(0.01–0.74)	0.02
Baseline PASI	1.01	(0.92–1.10)	0.89
Underlying diseases			
Erythrodermic PsO	2.33	(0.32–16.95)	0.40
PsA	1.05	(0.27–4.06)	0.95
HTN	0.45	(0.10–2.00)	0.29
DM	2.00	(0.32–12.40)	0.46
CVD	1.35	(0.07–25.47)	0.84
HBV	0.16	(0.01–2.34)	0.18
IGRA	2.29	(0.32–16.44)	0.41
Prior PsO treatments			
Acitretin	0.13	(0.01–1.38)	0.09
MTX	0.84	(0.11–6.45)	0.87
CsA	5.45	(0.91–32.75)	0.06
Phototherapy	6.78	(0.65–71.21)	0.11
Prior biologic agents	1.65	(0.25–11.01)	0.60
Diarrhea after injection	71.98	(9.31–556.47)	<0.001
PASI 75 at week 12	0.57	(0.14–2.29)	0.43

PsO, psoriasis; BMI, body mass index; PASI, psoriasis area and severity index; PsA, psoriatic arthritis; HTN, hypertension; DM, diabetes mellitus; CVD, cardiovascular disease; HBV, hepatitis B virus; IGRA, interferon-gamma release assay; MTX, methotrexate; CsA, cyclosporine.

## Data Availability

The data presented in this study are available on request from the corresponding author. The data are not publicly available due to privacy.

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
