# Peer review of "Risk Factors of Ixekizumab-Induced Injection Site Reactions in Patients with Psoriatic Diseases: Report from a Single Medical Center"

_biomedicines, 2023, doi:10.3390/biomedicines11061718_

Round 1

Reviewer 1 Report

In the retrospective study, the authors found the risk factors and the protective predictors for the occurrence of ixekizumab-induced ISRs in patients with psoriatic diseases.

Specific risk factors for ixekizumab-induced ISRs in patients with psoriatic diseases had not been well studied.

There are only a few information as follows. The incidence of ISRs decreases over time after repeating injections. Citric acid causes injection site pain.

The findings are the first and important to prepare for the ISRs and choose treatments.

Here are the comments.

1.Data are summarized in Table1, 2.

If characteristics are written in same line, and units of numbers are shown, we may be able to see Tables more easily.

2.If the incidences of ISRs on each dose and dosing interval of ixekizumab are available, the study may lead to slightly different results.

Author Response

Q1-1. Data are summarized in Table1, 2. If characteristics are written in same line, and units of numbers are shown, we may be able to see Tables more easily.

Response: Thank you for your suggestions and kind reminder. I have made the necessary revisions to the tables in the revised submission, including reorganizing the columns and rows, and adding the units of measurement for the numbers in the tables.

Q1-2. If the incidences of injection site reactions (ISRs) on each dose and dosing interval of ixekizumab are available, the study may lead to slightly different results.

Response: We appreciate the inclusion of the statement in the limitations section regarding the lack of analysis on the incidences of ISRs associated with each dose of ixekizumab and the relationship between dosing interval and ISRs incidence. This recognition emphasizes the importance of future studies to investigate these aspects and further enhance our understanding of potential dose-dependent effects.

Reviewer 2 Report

The article is very interesting and well written. The topic addressed by the authors is necessary to the therapeutic management of patients with psoriasis undergoing biologic treatment. The topic is very interesting, there is no such work in the literature so specific. Adverse reactions at the injection site are very common reactions, they are often confused with adverse reactions that lead to discontinuation of the drug instead real life experiences where it is established that they are manageable reactions are essential.

I have just a few revisions

1) The authors need to expand on the introduction or discussion part by also talking about what have been reported as effects related to ixekizumab treatment, such as vitiligo or other examples they can find in the literature, I leave two articles that may add something to the text

- DOI: 10.1111/dth.15102

- DOI: 10.1111/dth.15314

2) It is worth noting the authors' management of these patients who developed the reaction. I would recommend further discussion of the therapeutic management of these reactions as it would be an experience that could be shared with readers. Conclusion may be improved, I recommend rephrasing by talking about how the authors acted on the reactions encountered

3) Minimal revision of English is required

4) It would be preferable to add more photos if available to the authors

Minor editing of English language required

Author Response

Q2-1. The authors need to expand on the introduction or discussion part by also talking about what have been reported as effects related to ixekizumab treatment, such as vitiligo or other examples they can find in the literature.

Response: Thank you for your feedback. I have included the issue of common ixekizumab-related side effects and dermatological adverse effects in the introduction section, incorporating the references you provided.

Q2-2. It is worth noting the authors' management of these patients who developed the reaction. I would recommend further discussion of the therapeutic management of these reactions as it would be an experience that could be shared with readers. Conclusion may be improved, I recommend rephrasing by talking about how the authors acted on the reactions encountered.

Response: I have incorporated the issues related to the therapeutic management of injection site reactions (ISRs) in the introduction section as per your recommendation. Additionally, I have summarized these issues again in the conclusion section to reinforce their significance.

Q2-3. Minimal revision of English is required.

Response: Thank you for the reminder. I have had my manuscript reviewed by a colleague who is fluent in English writing.

Q2-4. It would be preferable to add more photos if available to the authors.

Response: I have included a series of three clinical photographs that capture the evolving nature of ISRs in another patient, providing visual insights into the temporal changes associated with the reaction.

Round 2

Reviewer 1 Report

No comments.

Reviewer 2 Report

Author's made the corrections